# Assessing Environmental Effects upon Modeling the Individual Lactation Curve of a Local Goat Population in a Pastoral System

**DOI:** 10.3390/ani14060942

**Published:** 2024-03-19

**Authors:** Ahlem Atoui, Maria Jesus Carabaño, Aicha Laroussi, Mouldi Abdennebi, Farah Ben Salem, Sghaier Najari

**Affiliations:** 1Laboratory of Livestock and Wildlife, Institute of Arid Regions (IRA), Medenine 4119, Tunisia; aicha.laarousi.25@gmail.com (A.L.); abdennebim@yahoo.fr (M.A.); farah.bensalem@ira.rnrt.tn (F.B.S.); najari.sghaier@ira.rnrt.tn (S.N.); 2Higher Institute of Human Sciences of Medenine, University of Gabes, Gabes 4100, Tunisia; 3Depto de Mejora Genética Animal, National Institute of Agricultural and Food Research and Technology (INIA), Ctra de La Coruña Km 7.5, 28040 Madrid, Spain; mjc@inia.csic.es

**Keywords:** local goat, lactation curve, pastoral system, herd management, Wood’s model

## Abstract

**Simple Summary:**

In pastoral farms, understanding the productivity of local goat milk is crucial; however, comprehensive data on complete lactation and the factors affecting lactation curves are limited. Our study aims to model local goat lactation curves and explore the factors affecting the parameters of individual lactation curves. The observed low production per goat and significant variation highlight potential opportunities for enhancement. Implementing improved farming practices and establishing a national genetic improvement program for a local goat breed in Tunisia could effectively address these challenges.

**Abstract:**

The present study aims to use Wood’s model to determine the parameters of individual lactation curves in a local goat population and their factors of variation under a pastoral system. A total of 137,927 records from 432 local goats were collected to assess the impact of litter size, year and month of kidding, herd and the age of the dam on lactation curve parameters. Wood’s model parameters were estimated using non-linear regression, and individual curves were fitted. The characteristics of the lactation curves were computed. The initial yield (A), rate of increase (B) and rate of decline (C) parameters in Wood’s model for local goats were 730 g, 0.26 and 0.09 respectively. The values of peak milk production (PP), peak date (PD) and persistency (PC) were 931.88 g/d, 23.39 days and 91.50%, respectively. Persistence was higher in goats with simple births, while peak production increased by around 0.3 kg for each additional kid. The curve parameters “A” and “C” differed according to the herd and month of kidding (*p* < 0.05); the age of the dam only had an impact on parameter “A” (*p* < 0.01). Parameter “B” was not significantly influenced by any of the factors considered (*p* > 0.05). Correlation coefficients among lactation curve characteristics were ranged from −0.20 to 0.89. Due to a significant negative correlation, selecting for parameter “A” may have an adverse effect on parameter “B”, resulting in a shorter time to reach peak production and less persistency, but an increase in peak production among goats. The curves derived from Wood’s model suggest that the shape of the curve may serve as a basis for herd management planning and to improve local population potentialities.

## 1. Introduction

In the majority of agropastoral systems, milk represents only secondary production [1]; the main livestock product is the meat of lambs and kids. Milk is rarely obtained due to the nomadic management of herds and the insufficient infrastructure of rangelands, which hinders the collection and commercialization of milk [2]. The milk performance of the local goat is considered low [3,4]; this is also observed for most goat populations raised in arid regions [5,6]. Najari et al. [7] suggest that reduced genetic dairy performances of local breeds are a result of a long natural selection favoring adaptation and regenerating capacity of autochthonous populations under harsh conditions. To achieve such evolution objective, natural and human selection reduce dairy performances, which has high energy to allow for genetic continuity with scarce resources. Moreover, a high dairy performance is assumed to have a negative genetic correlation with reproductive abilities. Milking, which is rarely practiced in pastoral systems, is practically limited to the peak of lactation period [7]. The consumption of fresh or processed milk is not a daily habit in pastoral society [8]. Nevertheless, good milking performance offers a relevant contribution to herd outcomes through the contribution to the kids’ growth and, thus, to meat production.

Dairy farmers can optimize their management decisions and selection processes by using lactation curves as useful tools for informed planning and strategy implementation. Lactation curve knowledge makes it possible to estimate the total production of milk from several test days early in the lactation process [1,9] or from a single test day [10]. Such information helps to make decisions about specific milk production early in the lactation process [11,12]. One specific objective is to identify goats that achieve peak production levels early in lactation and sustain these peaks throughout the entire lactation period. Essentially, the goal is to select goats characterized by greater persistence. Selecting goats with more persistent lactation is appealing for several reasons. Sustaining a relatively constant supply of milk throughout the year proves economically significant, particularly for dairy goats engaged in seasonal milk production. The physiological strain associated with negative energy balance in the initial stages of lactation could be mitigated, potentially resulting in a reduction in the occurrence and severity of metabolic and reproductive disorders. Additionally, maintaining a consistent diet for goats and increasing the proportion of roughage in their diet could offer cost saving benefits. The magnitude of the peak production and the ability to persist largely influence the overall shape of the lactation curve [13]. Nevertheless, there is a scarcity of research on factors influencing lactation curves in goats [14].

The lactation curve serves as a graphical representation, showing the relationship between milk production quantity and the duration of lactation [15]. Lombaard [16] characterizes it as a graph illustrating milk yield, initially rising at a relatively high rate until reaching peak production, then gradually declining until the end of lactation. The duration of lactation tends to be longer when the decline in milk yield is gradual and shorter when the decline is rapid. Animals with flatter lactation curves generally have longer lactation durations compared to those with curves showing rapid increases in daily milk production from parturition to peak, followed by a sharp decline.

Several non-linear models have been proposed to describe lactation curves, including those developed by Wood [17,18,19]. Additionally, multiphasic logistic models, such as the one by Grossman and Koops [20], and the Cappio-Borlinqo model [21,22,23,24] have been utilized. Notably, many studies have found that the Wood model adequately describes lactation milk yield in various dairy goat populations [25,26].

Milk yield and lactation curve shape are influenced by a multitude of factors, extensively investigated in animals subjected to both traditional management [27,28,29] and intensive management conditions [30]. Some factors are related to physiological traits of the goat, including lactation number [31,32,33], age at first kidding [34] and prolificacy [35]. Pregnancy itself has demonstrated an impact on milk production and lactation curve shape [35]. Environmental factors also play a role, with kidding in early spring associated with higher yields in goats under traditional management [27]. Additionally, the nutritional level during pregnancy, particularly in the last third of pregnancy, represents another environmental factor shown to influence milk productivity and lactation curves [36,37].

In dairy goats, the assessment of the lactation peak and other curve characteristics, such as the speed of reaching the peak and persistence, holds significant importance for both genetic and herd management considerations. It remains capital to establish specific local goat milking curves under aridity to characterize local genetic resources and their real performances under resource restrictions and irregularities under an ambulant herd management mode. Moreover, this step is essential to develop conceive appropriate zootechnical and genetic plans to preserve these genetic resources and to improve their efficiency as domestic livestock.

Under a pastoral system, a comprehensive understanding of the factors influencing milk productivity in the local goat population would be invaluable for farmers. However, there is scarce information available regarding complete lactation in local goats and the variables affecting lactation curves under pastoral management. Consequently, our study aims to model individual lactation curves of local goats under a pastoral system defined by harsh and irregular climatic conditions and explore factors influencing lactation curve parameters. The results would help to optimize the management of this genetic resource and its dairy performances, both direct and indirect milk production through kids’ suckling. Moreover, such information remains crucial for a possible genetic improvement of this population, taking into account the possible genetic interactions of milking performance with adaptation capacities.

## 2. Materials and Methods

### 2.1. Localization and Selection of Herds Subject to Milk Recording

The study was carried out on agropastoral herds raised on rangelands of the Tunisian arid zone delimited between the Great Eastern Erg and the Mediterranean coast. This region shares borders with the Gabes and Kebili governorates to the North, while its eastern boundary extends to Libya, and the western border abutting on Algeria. The zone is subdivided into four distinct natural regions: the mountainous region, characterized by the Matmata range mountains forming a vast valley in the South; the Dhaher “plateau”, situated to the West of the Matmata range mountains; the Oriental steppes of el Ouara and Jefara, positioned to the East of the governorate; and the large oriental Erg, a Saharan extension. The Mediterranean desert climate is marked by minimal annual precipitation (less than 100 mm) and elevated temperatures (more than 40 °C). In total, five breeders were selected and subjected to recording data.

### 2.2. Data and Animal Management

Milk records were collected between 1998 and 2001, from a dataset related to Tunisian local population under pastoral system. The Tunisian local goats are characterized by their small stature (Figure 1), with males averaging 76 cm and females 60 cm in height, as reported by Atoui et al. [38] and Ouni [39]. These animals were managed within an agropastoral and pastoral breeding system, with natural pasture serving as their primary food source. Notably, the quantity and quality of pasture exhibit substantial variations throughout the year. Pastoral transhumant herds move frequently on the wide area of collective rangelands to ensure the animals’ nutritive requirements under an irregular arid environment [38,40]. With the dry season, the quantity and quality of the pasture decrease, and supplemental feeding has to be provided. The primary mating period of local goats occurred between June and August. If a doe remained non-pregnant during the initial mating period, it was then moved to the group designated for mating during the subsequent period (October–November, corresponding to births in spring). The kidding season started in October and extended until February, with a concentration of births in November and December. On average, the suckling period lasted for 120 days. Female kids were initially mated between 12 and 18 months, depending on their season of birth and their body conditions.

Milk production was collected every 14 days. Kids were separated from their dams at approximately 18:00 h on the evening preceding the milking day. The total quantity of milk collected on the test day was considered as the morning daily production, reflecting a once-a-day milking routine for the goat. To estimate the daily milk production for each goat, it was assumed that the actual daily production could be achieved if the animals were milked twice a day. Information regarding the herd, the date of kidding, the morphological type and the litter size of goats was documented. The quantity of milk available per goat per test day was measured using graduated plastic cups of 1000 mL capacity. The dataset contained 137,927 milk records from 432 local goats. The distribution of recorded lactations is presented in Table 1. Approximately 66% of parities were single, and only about 34% were twin. The highest number of deliveries in local goats occurred between 1999 and 2000.

### 2.3. Statistical Analysis

The data collected under pastoral farming conditions was subjected to several steps of verification and elaboration to fix the suitable model to assess the local goat lactation curve; we tested the majority of the cited functions in the bibliography for the adjustment of the lactation curves of sheep and goats. As for the estimation of the curve parameters and the numerical criteria for evaluating the models applied, we applied a non-linear regression procedure, and the model resolution was perfected by an iterative procedure [41]. Table 2 groups the formulas of the applied models with the results of the iterative process applied for parameter estimation.

Based on these results, the most appropriate model for our case is Wood’s. So, we opted for this function to estimate the lactation parameters and the curve shape of the local goat. Then, the database was analyzed using the statistical software SPSS 25.0 (IBM SPSS Statistics for Windows, Version 25.0. IBM Corp., Armonk, NY, USA). Descriptive statistics including mean and standard deviation (SD) were calculated for each variable. The lactation curve, adjusted using the Wood function as proposed by Rekik et al. [29], is defined as follows:Y_t_ = At^B^ exp^−Ct^(1)
where Y_t_ represents dairy production in kilograms, and t denotes the number of days post parturition at milk recording.

The parameters of the Gamma curve (A, B, and C) offer valuable insights into various dairy parameters:A represents initial dairy production (Y0).

B is the rate of increase to peak production; C is the rate of decline after peak production; and “exp” is the exponential.

2.The date of peak production (in days) is calculated as B/C.3.The production at the peak (in kilograms) is determined by the formula (A × (B/C)^B^) × exp^−B^.4.The coefficient of persistence (%) is calculated as 100 − (−B + 1) × ln(C).

A non-linear procedure was applied to solve, by an iterative procedure, the model to estimate the Equation (1) parameters.

5.Total production: the integral of the function Y_t_ = At^B^ exp^−Ct^ between the first day of kidding and the date of the last recording +7.6.Lactation length: the difference between the date of last recording and the date of kidding +7.

In addition to estimating the overall curve, individual parameters A, B, and C, considered as quantitative variables or phenotypes for each goat, and their relative overall variabilities were assessed by solving a model in which the non-genetic variation factor assumed to have an effect on milking ability was included.

After normality checking via Kolmogorov–Smirnov test (*p* < 0.05), two-way analysis of variance (ANOVA) was performed to determine the effects of the year of kidding, the month of kidding, the age of the dam at kidding, the herd and the litter size on the lactation curve parameters. Duncan’s multiple range tests were used to determine the significant differences among the means at a probability level of *p* < 0.05.

The adopted model was as follows:Y_ijklmn_ = μ + H_i_ + Z_j_ + M_k_ + L_l_ + A_m_ + e_ijklmn_(2)

Y_ijklmn_ represents the estimated curve parameters; μ is the overall mean; H_i_ denotes the herd effect (i = 1, 2, 3, 4 and 5); Z_j_ represents the year of kidding effect (j = 1998, 1999, 2000); M_k_ denotes the month of kidding effect (k = January, February, November, December); L_l_ signifies the litter size at kidding effect (l = simple, double); A_m_ reflects the age of dam at kidding effect (m = 1, 2, … 8) and e_ijklmn_ denotes the residual error.

Pearson’s correlation coefficient was employed to evaluate the relationship between lactation curve characteristics.

## 3. Results

Figure 2 illustrates the estimated lactation curve in this local goat population under a pastoral system. The lactation curves of this population exhibited a pattern where milk production gradually increased during early lactation, reaching a peak approximately 2 to 5 weeks postpartum, followed by a gradual decline until the end of lactation period. Note that the variability observed in the data seems important between 30 and 100 days of lactation.

Table 3 provides descriptive statistics of lactation characteristics in the local goat population under a pastoral system.

The initial production is 730 g/day. The estimated persistence coefficient was 91.5%. The “B” parameter of the local goats was 0.26 ± 0.04. The “C” parameter was 0.09 ± 0.01.

The effect of various factors on the estimates of lactation curve parameters for a pastoral system is detailed in Table 4. The coefficient of determination R^2^ varied between 0.81 and 0.90 across traits.

A notably high R^2^ reflects the significance of the fixed effects incorporated into the adopted model. Effects related to environmental and management conditions over time (year and month of kidding) demonstrated high significance on all studied parameters (*p* ˂ 0.01). However, effects related to groups of dams, such as the age of the dam at kidding, exhibited a relatively modest level of significance overall (*p* < 0.05). As anticipated, the herd factor exerted the most substantial influence on the total production. Conversely, the litter size displayed a less significant effect on the total production, with its greatest impact observed on lactation length.

The means comparison test results, for each significant non-genetic factor, are shown in Table 5. Duncan’s test reveals a significant variation in lactation curve parameters across the herds. There are notable differences in milk production between the goats of herds 2, 3, 4 and 5, compared to those recorded in herd 1. Herd 1 achieved a significantly higher total production. Additionally, herd 2 demonstrated a significantly extended lactation period. However, herd 4 exhibited comparatively lower daily production levels than the others.

Milk production varied across different months, with the lowest initial production occurring in February (280.84 g/j) and the highest in December (807.80 g/j). The peak production of 1.23 kg was observed in November, later than the other months. January followed with a peak of 832.41 on day 37. In terms of persistence values, February kidding exhibited higher rates (94.10%), whereas November kidding displayed comparatively lower values (93.30%). During, November kidding, goats initiated with a higher average production and promptly reached a production peak.

In our study, as depicted in Table 4, the lowest level of production was observed in young goats, while older goats also demonstrated suboptimal production despite an extended lactation period. For peak production and peak date, it becomes evident that younger goats achieve higher peak productions (1420.52, age = 3 years) and reach their peak earlier (35.88, age = 3 years) compared to older goats. Older goats present a higher lactation length compared to younger goats.

Litter size had a relevant effect on all parameters. The estimated values for lactation curve parameters for simple kidding were usually were lower than for twin kidding (Table 4). The total production showed a difference of 80 kg between simple and twin pregnant goats. The mean daily production was 0.92 and 0.66 kg/day, while the average lactation length was 167.26 and 138.73 days for twin and simple pregnant goats, respectively. Simple births showed the lowest initial production (536.85 g), only slightly greater than twin births (557.26 g). In terms of production in the peak and the day on which the peak is reached, it is observed that goats that had a twin pregnancy produced the most at the peak (689.87 g), and that the peak was reached earlier (22 days) than for the simple pregnant goats.

Table 6 shows the correlations between the studied traits. The correlation between the parameters «A» and «B» was moderately negative (−0.25), whereas the correlation between the parameters «A» and «C» (0.15) was not very strong. The parameters «A» and peak production and «A» and peak date had correlations of 0.80 and −0.15, respectively. The parameters A, B, C and the persistence coefficients had a negative correlation. Peak date showed the strongest correlation with peak production (0.86) but only moderate correlation with total production (0.40) and the persistency coefficient (0.57). Total production showed a moderately high correlation with persistency (0.62) and a higher correlation with lactation length (0.71), as might have been expected.

## 4. Discussion

The lactation curve of the local goat population exhibited a relatively plateaued shape. Milk production increased until it reached a peak before the 24th day of lactation, followed by a gradual decrease until the end of lactation length. On average, the observed initial production was 730 g, and the peak of lactation corresponded to an average production of 932 g. However, this average performance varied widely across individuals, indicating significant variability within the population. Additionally, other researchers have observed a similar pattern to that depicted in the lactation curve characteristics of local goats [30] and sheep [31]. The observed trend is also consistent with prior research findings [14] and [6], suggesting that goat lactation curves exhibit a stable profile compared to those of cattle.

The parameters of the lactation curves of the local goat population, as estimated by the Wood model, are quite comparable to those observed in suckling goats raised in arid environments [42,43,44,45]. In fact, dairy performances of local goats illustrate similar productive abilities to those characterizing authentic breeds and populations raised under arid and desertic environments [8]. Note that their small weights and reduced milk production performances reduce the animals’ requirements, allowing goat mothers to survive and to feed their kids with reduced pastoral and irregular forage resources under such a harsh environment. This genetic profile remains one of principal adaptation criteria. The total milk production values obtained in this study were comparable to those observed by [14] for exotic dairy goats under research station conditions in Kenya, but were higher than those reported by [16] for the same breed raised in tropical environments.

The lactation peak occured on the 24th day, which appears relatively early compared to the more dairy specialized Murciana breed, where the peak was observed on the 35th day [46]. The initial production was 730 g, reaching a maximum of 960 g, which remains low compared to the Murciana breed with 1880 g [46]. The result was lower than the 1290 g obtained by Ayasrah et al. [47] in Damascus goats. The total estimated production of approximately 980 g confirms the results obtained by Jalouali [3]. Although it was higher than that of East African and Galla goats [48], and Nigerian Sahel and West African Dwarf goats [49], it is in accordance with the findings of Osinowo and Abubakar [50]. According to previous research on tropical goats [49,51,52,53], the peak production found in this study is consistent with those findings. Furthermore, the observed peak day is consistent with the range documented for sheep [54,55] and tropical goats [48]. The local goat showed a modest level of persistency (91.50%) during the 120-day lactation period, indicating their moderate capacity to maintain their milk availability. The estimate is lower than 128%, which is that of a single lactating Friesian cow, determined randomly from 24 h of production on one day every week during a period of 305 days [32]. The persistency value for local goats was higher than 60.17, found by Waheed and Kahn [56] in Beetal goats, and those published for goats from East Africa and their hybrids (2.87 to 4.50%) with the same lactation time of 120 days [48].

The coefficient of determination, R^2^, exhibited a high value, indicating the importance of fixed effects included in the model. The current values were consistent with those reported in the existing literature [14,57], albeit slightly differing from others [6,14]. Cigdem et al. [58] report a lower R^2^ of 0.66 and 0.63 for Bornova and Saanen goats, respectively. Waheed and Kahn [56] reported a higher R^2^ value of 0.98 for Beetal goats.

The herd significantly influenced the characteristics of the lactation curve in the local goat population. Management and within-herd improvement might explain the herd influence on the local goats’ lactation curve. While daily grazing is a characteristic of the pastoral south Mediterranean small ruminant herd, the type and timing of the animals’ food supplementing varies largely throughout herds in relation to relative resources and constraints. Furthermore, the herders’ pastoralist practice, over centuries and since nomadism time, with a certain degree of selection or profile choice, has been practiced based on the information that each farmer has gained via personal experience or ancestry, which causes subsequent variations in the production of the herds. Herd 1 is well recognized in the region for its high milk potential. The impressive performance of this herd is not solely attributed to animal management but primarily to the impact of meticulous, selective breeding by the farmers generations. Since the establishment of the herd in 1950, the farmer has consistently emphasized selection for milk production. The depth of the farmer’s knowledge regarding the individual attitude of the goats and their pedigrees reflects the precision of the approach, resulting in the creation of a herd distinguished by the milk characteristics of their goats. The total production of herd 1 aligns with the findings in the literature concerning this population under oasis conditions [3,4].

The kidding month factor showed a significant effect on the lactation curve parameters, with most goats exhibiting a seasonal breeding pattern, experiencing estrus in late summer–autumn and giving birth in the winter–early spring period. The month of kidding has been observed to influence both initial and peak production as well as persistency in lactation. According to the results mentioned in Table 4, goats kidding in December demonstrated lower initial and peak production compared to those kidding in November. Despite the lower initial and peak production, the total production in December was greater for goats kidding early than for those obtained kidding in the other months, similar to the findings of [59]. Kamel [60] and Hamed [61] observed a significant effect of the month of kidding on the initial milk yield, respectively, in Shami goats and Zaraibi goats. The influence of the kidding month on the lactation length derives from distinct phases. Initially, goats who give birth early (November and December) demonstrate a longer lactation length than those kidding later. In England, goats kidding between February and May attain peak yield between 9 to 12 weeks of lactation, whereas those kidding from June to January reach peak yield earlier, around 5 to 8 weeks of lactation [59]. Additionally, those goats kidding between February and May exhibit greater initial yields and persistence than those kidding between June and January. However, Ronningen [62] reported conflicting results in Norway, where goats kidding from December to February displayed more persistence and peaked earlier than those kidding in March or April. Takma et al. [6] reported different shapes of lactation for different seasons of kidding, which may be due to constraints in feed availability. Catillo et al. [17] and Dhanoa [18] concluded that a clear separation cannot be found among lactation curves pertaining to different seasons of birth. The effect of the season of kidding is largely dependent on the availability of green fodder and management practices adopted to protect the animals against adverse climatic conditions [63].

Similar effects of the age of the dam at kidding on lactation curves have been documented in East African goats, as reported by Ronningen [63]. The class of age 3 years exhibited the highest production, followed by ages 4 and 5 years, while goats of age 2 years had the lowest production level. Production peaks for ages 2, 3 and 4 years were higher than those for ages 7 and 8 years. Persistency, referring to the goat’s capacity to sustain peak milk production, exhibited its highest levels in younger goats and diminished as age increased [7,9]. In our study, younger goats had the highest persistence coefficient. The lowest persistence coefficient was obtained for goats aged 8 years. Similar to our studies in dairy cattle [8], examined persistency in dairy goats. They found that the persistence coefficient was 91% for goats aged 3 years, decreasing to 84% for goat aged 5 years. Generally, the age of the dam significantly influenced the lactation curve, with the initial and peak yields increasing until the age of 5 years and then declining. The time of peak production occurred earlier, and persistency decreased with increasing age [6]. The findings of the current study align with those of Hamed [61] concerning Zaraibi goats and those of Akpa et al. [64] regarding Red Sokoto goats. However, they do not coincide with the results reported by Kamel [60] in Shami goats, who concluded that the age of the dam had no significant influence on the initial milk yield. The current study’s results also correspond with the findings of Ayasrah et al. [47] in Damascus goats.

Significant changes in lactation curve parameters have been attributed to litter size. In fact, selection along with appropriate reproductive control may be the cause of the capacity for weaning more kids. Higher dairy production levels for goats with double births were seen in the current study, which is consistent with findings from other authors such Cannas et al. [37] and Dag et al. [13]. This implies that litter size and milk production are positively correlated. This might be caused by an increased placenta volume in animals with several kids, which, in turn, leads to an increased synthesis of placental lactogen, the hormone principally responsible for the development of the udder in mammals. The findings regarding the initial milk yield “A” were consistent with those reported by Rojo et al. [65] for Alpine, Saanen, and Anglo-Nubian goats. Ayasrah et al. [47] observed no significant impact of litter size on initial milk yield in Damascus goats. However, these results were contradictory to earlier studies conducted by Kamel [60] in Shami goats, Hamed [61] in Zaraibi goats, and Akpa et al. [64] in Red Sokoto goats, which reported different outcomes.

The higher initial milk yield observed with twin kids may be attributed to a greater stimulation effect on the mammary gland, as multiple kids suckling on the udder provides increased stimulation (Salvador and Martínez [66]). Does with single kids exhibited a higher rate of milk production, rising (“B”) before reaching peak production compared to does with twin kids. As litter size increased, the rate of milk production ascent to peak decreased. Ayasrah et al. [47] did not find a significant effect of litter size on the rate of milk production ascent to peak in Damascus goats. However, these findings were inconsistent with previous reports by Kamel [64] in Shami goats, Hamed [61] in Zaraibi goats, and Akpa et al. [64] in Red Sokoto goats.

This suggests that twin litter size could slow down the rate of milk production ascent to peak yield due to the higher number of suckling sessions, which evacuate milk from the alveoli. Consequently, it takes a longer time for the secretory milk gland to reach peak milk production.

Selection for parameter “A” could potentially have adverse effects on parameter “B”, leading to a reduction in the time required to reach peak yield and a decrease in persistency, but with an increase in peak yield among local goats due to a notable negative correlation. These results are in agreement with those of Abubakr [67] and Hamed [61]. The correlation between parameters “A” and “C” was relatively lower (r = 0.15), contrasting with the highest correlation previously reported by Gipson and Grossman [53] (r = 0.46). Moreover, Devendra [52] highlighted a maximum correlation of 0.37 between “A” and any shape parameters. The strong correlation between “B” and “C” (r = 0.92) aligns with the findings of Cannas et al. [37], who found a value of 0.87. In our study, the correlation between “B” and “C” was notably higher (r = 0.92), surpassing the correlation reported by Pollott et al. [30].

## 5. Conclusions

Data collected through milk control in pastoral farms were used to assess the milk potential of a local goat population by analyzing global and individual lactation curve shape and parameters using Wood’s model. Daily production reveals moderate milk capacity, consistent with numerous local goat breeds in arid and desertic regions. The curves derived from the Wood model, influenced by the factors considered in this study, offer valuable insights for effective herd management within the pastoral system. Individual lactation curve analysis in goats reveals that arid region conditions influence local population production potential. Environmental factors, such as kidding year, significantly impact lactation curve shape due to limited forage resources. These insights can be utilized for strategic planning, including adjustments in culling practices and the evaluation of nutritional and health conditions in animals within pastoral contexts. This study highlights that the dairy production of local goats and agropastoral breeding modes can be improved to consolidate their resilience by reasonable programs of selection and herd management. Instead of meat production, this sector can consider milk production on the selection objective list. Moreover, these findings may contribute to the creation of impartial methodologies for comparing animals with incomplete lactation records, particularly in the context of genetic evaluation.

## Figures and Tables

**Figure 1 animals-14-00942-f001:**
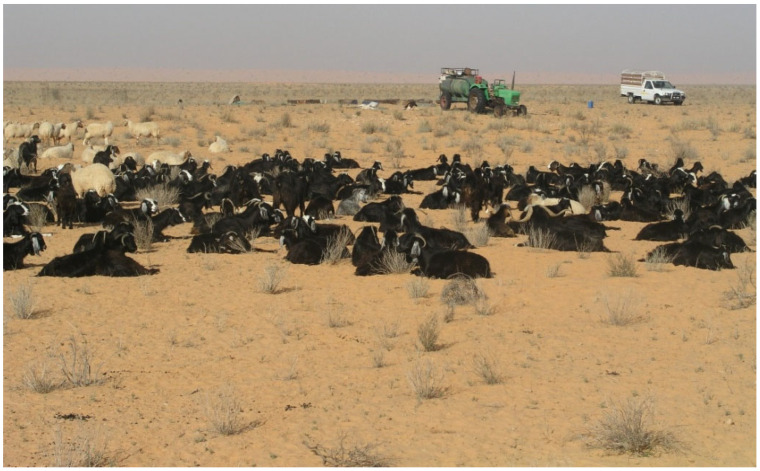
Local goat population in pastoral system.

**Figure 2 animals-14-00942-f002:**
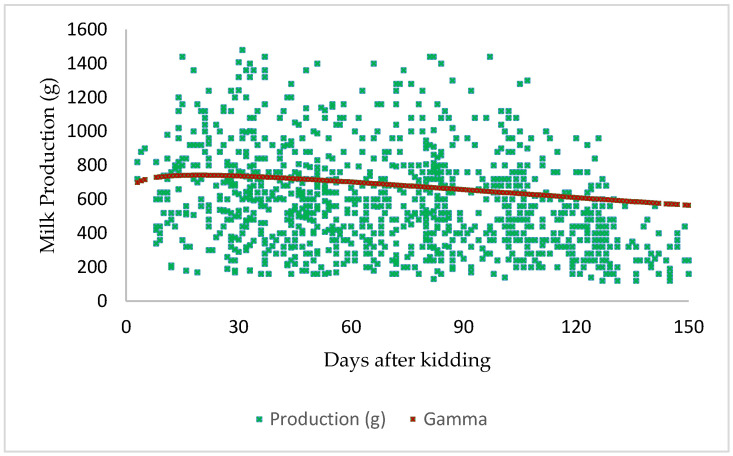
Lactation curve of local goat in pastoral system.

**Table 1 animals-14-00942-t001:** Distribution of recorded lactations in pastoral system.

Herd	Kidding Mode		Year			Total
	Single	Twin	1998/1999	1999/2000	2000/2001	
1	96	-	-	96	-	96
2	108	-	-	108	-	108
3	600	370	286	355	333	974
4	468	102	148	-	-	148
5	54	27	-	66	0	66
	904	488	434	625	333	1392

**Table 2 animals-14-00942-t002:** Comparison criteria of models used for describing lactation curves of local goats under a pastoral system.

Models	Equation	N.it	R^2^	RSS
Wood	A × (days × B) × exp(−C × days)	13	0.77	7,632,084
Yaday	A + (B/days) + C × days	16	0.75	8,021,316
Dhaona	A × (exp(−B × (days − C))) × exp(−2 × exp(−D × (days − C)))	40	0.45	8,225,559
Wilmink	A + B × exp(−D × days) + C × days	56	0.39	9,878,568
Capio-Borlino	A × days × (B × exp(−C × days))	18	0.65	8,197,684
Cobby et Le Du	A × (1 − exp (−B × days) × exp(−C × days))	-	-	-
Morant	A × exp(B × ((days − 150)/100) + C × (((days 150)/100) × 2) + (D/days))	22	0.66	8,065,465
Goodall	A × (days × B) × exp(−C × days + D)	24	0.61	8,632,084
Grossman	A × (days × B) × (exp(−C × days)) × (1 + D × sin(days) + e × cos(days))	29	0.58	8,003,331

A, B, C and D: curve parameters. N.it: number of iterations required for convergence. R^2^: coefficient of determination. RSS: residual sum of squares. days: days after kidding.

**Table 3 animals-14-00942-t003:** Descriptive statistics for lactation curve characteristics estimated by the Wood model in a pastoral system.

Variables	Mean	Min	Max	SD	Normality Test
Initial production (g/day)	730.00	150.00	960.00	250.00	Accepeted
Rate of increase to peak	0.26	0.10	0.30	0.04	Accepeted
Rate of decline after peak	0.09	0.05	0.12	0.01	Rejected
Lactation length (days)	146.00	62.00	189.00	30.71	Accepeted
Total production (kg)	97.97	17.08	320.10	87.63	Accepeted
Mean production (kg/day)	0.66	0.12	1.77	0.44	Accepeted
Peak date (days)	23.39	4.48	81.67	36.12	Rejected
Persistence coefficient (%)	91.50	76.98	96.70	96.60	Accepeted
Peak production (g/day)	931.88	121.05	5189.40	845.80	Accepeted

Min: minimum, Max: maximum, SD: standard deviation.

**Table 4 animals-14-00942-t004:** ANOVA significance test and coefficient of determination (R^2^) for a model incorporating non-genetic factors on lactation curve parameters.

Factors	DF	A	B	C	LL	MP	TP	PD	PC	PP
Herd	4	S	S	S	S	S	HS	S	S	S
Year of kidding	2	HS	HS	HS	HS	HS	HS	HS	HS	HS
Month of kidding	3	HS	HS	HS	HS	HS	HS	HS	HS	HS
Litter size at birth	1	S	S	S	HS	S	S	S	S	S
Age of dam	8	S	S	S	S	S	S	S	S	S
R^2^	-	0.83	0.80	0.80	0.90	0.89	0.87	0.87	0.81	0.82

DF: degrees of freedom; S: significant (*p* < 0.05); HS: significant (*p* < 0.01); A: initial production; B: rate of increase to peak; C: rate of decline after peak; LL: lactation length; TP: total production; MP: mean production; PD: peak date; PC: persistence coefficient; PP: peak production.

**Table 5 animals-14-00942-t005:** Variation of lactation curve parameters based on fixed factors.

Factor	N	A	B	C	LL	MP	TP	PD	PC	PP
**Herd**										
1	26	243.21 ^b^	0.24 ^b^	0.09 ^c^	199.61 ^a^	1.63 ^a^	318.51 ^a^	57.01 ^c^	92.50 ^c^	2091.34 ^a^
2	16	99.668 ^c^	0.21 ^c^	0.06 ^d^	138.46 ^d^	0.31 ^c^	45.54 ^c^	55.49 ^c^	93.10 ^b^	327.09 ^d^
3	16	17.75 ^d^	0.11 ^d^	0.10 ^b^	185.31 ^b^	0.35 ^c^	57.84 ^c^	68.88 ^b^	91.00 ^d^	426.65 ^c^
4	15	17.95 ^d^	0.09 ^d^	0.05 ^d^	178.27 ^c^	0.25 ^c^	45.06 ^c^	83.38 ^a^	90.00 ^e^	306.03 ^e^
5	144	449.95 ^a^	0.28 ^a^	0.11 ^a^	141.93 ^e^	0.62 ^b^	86.44 ^b^	30.00 ^d^	94.00 ^a^	739.66 ^b^
**Year of kidding**										
1998	64	417.35 ^b^	0.22 ^b^	0.08 ^b^	154.84 ^b^	1.12 ^a^	185.13 ^a^	74.75 ^a^	92.80 ^c^	1124.23 ^a^
1999	102	153.03 ^c^	0.12 ^c^	0.05 ^c^	161.59 ^a^	0.57 ^c^	78.52 ^b^	44.90 ^b^	93.30 ^b^	539.05 ^c^
2000	51	548.08 ^a^	0.27 ^a^	0.10 ^a^	137.11 ^c^	0.43 ^b^	70.14 ^b^	18.85 ^c^	94.40 ^a^	707.75 ^b^
**Month of kidding**										
January	32	429.43 ^b^	0.20 ^b^	0.06 ^c^	130.78 ^c^	0.71 ^a^	93.2 ^a^	36.81 ^b^	93.80 ^c^	832.41 ^b^
February	24	280.84 ^d^	0.08 ^c^	0.05 ^c^	108.65 ^d^	0.75 ^a^	82.06 ^a^	27.43 ^c^	94.10 ^a^	774.88 ^c^
November	5	807.80 ^a^	0.25 ^a^	0.10 ^a^	178.5 ^a^	0.28 ^b^	48.97 ^b^	51.78 ^a^	93.30 ^d^	1238.49 ^a^
December	157	300.14 ^c^	0.23 ^a^	0.09 ^b^	164.266 ^b^	0.67 ^a^	114.38 ^a^	22.00 ^d^	93.90 ^b^	352.57 ^d^
**Litter size**										
Simple	139	536.85 ^b^	0.27 ^b^	0.08 ^b^	138.73 ^b^	0.66 ^b^	85.79 ^b^	21.95 ^b^	90.11 ^b^	357.81 ^b^
Twin	76	557.26 ^a^	0.22 ^a^	0.10 ^a^	167.14 ^a^	0.92 ^a^	166.1 ^a^	22.48 ^a^	93.10 ^a^	689.87 ^a^
**Age of dam**										
1	4	430.43 ^c^	0.22 ^a^	0.08 ^b^	90.5 ^d^	0.64 ^bc^	51.99 ^d^	47.01 ^c^	94.50 ^a^	532.66 ^e^
2	28	260.84 ^e^	0.09 ^d^	0.05 ^c^	145.42 ^c^	0.54 ^c^	83.31 ^bcd^	45.49 ^c^	93.40 ^b^	760.88 ^e^
3	60	807.808 ^a^	0.26 ^a^	0.09 ^b^	160.43 ^ba^	0.68 ^abc^	117.69 ^b^	35.88 ^f^	93.20 ^c^	1420.52 ^a^
4	22	500.14 ^b^	0.24 ^a^	0.10 ^a^	147.59 ^c^	0.51 ^c^	73.97 ^cd^	53.38 ^a^	92.78 ^c^	1200,23 ^b^
5	31	440.49 ^c^	0.23 ^a^	0.07 ^c^	142.46 ^c^	0.81 ^ab^	110.57 ^bc^	40.00 ^e^	92.30 ^d^	970.56 ^c^
6	27	400.04 ^d^	0.21 ^b^	0.06 ^c^	167.03 ^a^	0.89 ^a^	156.67 ^a^	40.22 ^e^	92.10 ^c^	823.56 ^c^
7	30	365.23 ^e^	0.10 ^c^	0.05 ^c^	155.63 ^b^	0.58 ^bc^	91.86 ^bcd^	41.21 ^d^	92.00 ^e^	777.56 ^d^
8	12	300.03 ^e^	0.08 ^d^	0.05 ^c^	165 ^a^	0.56 ^bc^	91.19 ^bcd^	50.08 ^b^	91.30 ^e^	650.23 ^e^

^a, b, c, d, e, f^ Means with different superscripts within a column differ significantly with respect to the same parameter (*p* < 0.01). A: initial production; B: rate of increase to peak; C: rate of decline after peak; LL: lactation length; TP: total production; MP: mean production; PD: peak date; PC: persistence coefficient; PP: peak production.

**Table 6 animals-14-00942-t006:** Correlation coefficients between lactation curve parameters in local goat population under a pastoral system.

	A	B	C	LL	TP	MP	PD	PC	PP
A	1	−0.25 **	0.15 *	0.20 *	0.82	0.80	−0.15	−0.26	−0.38
B		1	0.89	0.15	0.55	0.46 **	0.42	−0.35	0.60 *
C			1	0.03	0.12	0.22	0.31	−0.25	0.50
LL				1	0.71	0.60	0.11 *	0.46	−0.20
TP					1	0.59	0.40	0.62	0.46
MP						1	0.51	0.43 **	0.39
PD							1	0.57	0.86
PC								1	0.87 **
PP									1

** *p* < 0.01 * *p* < 0.05. A: initial production; B: rate of increase to peak; C: rate of decline after peak; LL: lactation length; TP: total production; MP: mean production; PD: peak date; PC: persistence coefficient; PP: peak production.

## Data Availability

The data presented in this study are available on request from the corresponding author. The data are not publicly available due to privacy and ethical restrictions.

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
