# Peer review of "Assessing Environmental Effects upon Modeling the Individual Lactation Curve of a Local Goat Population in a Pastoral System"

_animals, 2024, doi:10.3390/ani14060942_

Round 1

Reviewer 1 Report

Comments and Suggestions for Authors

I am writing the review for the manuscript entitled 'Assessing Environmental Effects upon Modeling Individual Lactation Curve of Local Goats Population in Pastoral System' submitted to mdpi Animals. 

In this manuscript, the researchers assess the lactation curves parameters in a population of local goats in Tunisia. 

The research is extremely interesting and the proposed approach is adequate. However, there are major language flaws that must be addressed before a better review. Additionally, the methodology needs major improvement both in the description as in the justification.

Major comments:

The paper must go through an extensive language review. Some parts of the paper are hard to follow. There are typos, grammatical errors, and structural erros. A list if provided, but it is not exhaustive. 

Methodology: The statistical analysis has a few major issues:

a) It seems that a single lactation curve is described in equation 1 (lines 142/143). However, some parts of the paper appear to evaluate many different curves, which is not described. 

b) Figure 3 includes a single lactation curve which is extremely noisy. That indicates the need to identify the sources of noise in the curve. Additionally, it seems that the lower limit is below 0. Please, change the y-axis.

c) I would recommend fitting different lactation curves for the different factors. If that was done, it must be explicit in the material and methods. 

d) Only one formula for the lactation curve was described, which is fine. However, the only reference regarding that recommendation is from dairy cattle. There are many research papers in lactation curves in goats, which must be incorporated in this research. 

Specific comments:

The abstract must be rewritten, especially the results. The use of parameters A, B, and C is not clear unless the reader has access to the formula. Please, focus on the major picture. 

Introduction:

-Must be reviewed before specific comments

Material and Methods

-Must be reviewed before final comments.

-Please, see the major comments.

-Rewrite lines 147 to 159 in paragraphs. 

-Explain why 7 was added to the lactation last recording 

Results and Discussion

-Literature seems to be accurate. However, there are many papers about dairy goat lactation curves that were not cited. While they usually focus on major breeds, it is still a better comparison than Holstein cows (reference 20).

-Grammar must be reviewed before specific comments

Conclusion:

-Your paper's aim was "to model individual lactation curves of local goats under pastoral conditions and 85 explore factors influencing lactation curve parameters". However, you didn't explore the parameter as much as you could, as a single curve was fitted. Finally, the conclusion goes over many topics that were not included in the aims. Please, readjust either the aims or the conclusion.   

Grammatical-

This is not an exclusive list:

· Line 44 - hole herd, , that line is not clear,

· Line 78 - essentaul tu

· Line 79 - her efficiency

· Line 98 - ecologically definite arid

· Line 101 - 5 breeders

· Line 109 - Pastoral breeding mode

· Line 113 - Not clear

· Line 133 - capacite

· Line 139 - database was analysed(space)

· Line 193 - evolutedun der arid

· Line 194 - leavy weights??

· Line 197 - SD 87.63

· Line 274 - ptoductive

· Line 275 - evolutedun der

· Line 276 -leavy

· Line 300 - both on the curve shape, target and parameters

· Line 339 -gots

· Line 370 - selecting(select)

· Line 376 - improve

· Line 377 -rather

· Line 378 - abjective

Comments on the Quality of English Language

English must be reviewed. There are many typos, grammatical errors, and structural errors. 

Author Response

Dear Editor,

Thank you for your communication regarding our manuscript. We appreciate the opportunity for reconsideration and have diligently addressed all the requested revisions as highlighted in the revised manuscript.

The modifications made to the manuscript are clearly marked for your convenience. Furthermore, we have provided detailed responses to each of the reviewers' comments, addressing their points systematically. We believe that these revisions significantly enhance the quality and clarity of the manuscript, aligning it with the standards of animals journal.

We look forward to your assessment of the revised manuscript and appreciate your continued guidance throughout this process.

Thank you for your consideration.

Reviewer 2 Report

Comments and Suggestions for Authors

In the manuscript, the authors evaluated environmental impacts by modeling the individual lactation curve of the local goat population in the pastoral system. The subject of this study is suitable for the “Animals” journal. The authors claim that the curve derived from the gamma-type Wood’s model may serve as the basis for herd management planning and adjustment. The study is well designed and presented, but I offer a few corrections to increase the scientific value of the manuscript. After the authors address these corrections, the manuscript can be accepted after revision.

Also;

-The results are convincing and supported by the discussion.

-The conclusions are consistent with the evidence and arguments.

-The topic and references are appropriate.

-Tables and figures are presented clearly and understandably.

-Line 114: Please use a different word instead of “regression”.

-Table 1: Please use “twin” instead of “double”. Also, why the herd numbers not in consecutive? (3, 4, 8, 1, 2)

-Line 149: “e is the exponential” should be “e is the natural log base (2.7182)”

-Line 164: Why did you use “Shapiro-Wilk” for normality? Your sample size is over than 50 please use Kolmogorov-Smirnov one sample test for normality.

-Why you chose Wood’s model instead of other models, please specify.

The “Introduction” section is well but please more specify your aim.

The “Materials and Methods section” can be acceptable but please make necessary corrections.

The “Result” section is enough.

The “Discussion” section was supported by the results and literature and looking good.

The “Conclusion” section is OK, but you should mentioned how you can say these for today because your data is from 24 years ago?

Author Response

(The authors gave the same response as above.)

Reviewer 3 Report

Comments and Suggestions for Authors

Assessing Environmental Effects upon Modeling Individual Lactation Curve of Local Goats Population in Pastoral System

The primary objective of this study was to evaluate the milk potential of local goats in pastoral systems by analyzing individual lactation curves and global lactation curve shapes using Wood's model or the Gamma function. The study aimed to explore factors influencing milk production in local goat populations, including the age of the dam, litter size, and other environmental variables. By scrutinizing lactation curves and associated parameters, the study aimed to offer insights for effective herd management within pastoral systems, encompassing strategic planning, culling practices, and the assessment of the nutritional and health conditions of the animals. Additionally, the study aimed to contribute to the enhancement of dairy production in local goats through selection and herd management programs, underlining the significance of milk production in pastoral contexts.

The main issues identified in this paper are as follows:

1. Limited Genetic Information: The study lacks details regarding the genetic background of the animals, specifically the breed or breeds involved. Providing a comprehensive overview of the local breeds could enhance the manuscript and facilitate comparison and discussion of the results with other breeds.

2. Statistical Analysis: Although the study employs statistical analyses such as ANOVA and Pearson's correlation coefficient to evaluate the effects of various factors on lactation curve parameters, there may be limitations in the methodology or interpretation of results. For instance, while the study acknowledges the influence of factors such as the age of the dam and litter size on curve parameters, further elaboration on the extent of these effects and their practical implications would be beneficial. One suggestion is to explore the potential interaction of factors by creating contemporaneous groups (e.g., herd, month of kidding, year of kidding).

3. Language Review: Conducting a thorough language review would enhance the manuscript's clarity and coherence.

Addressing these limitations, perhaps by discussing them in a dedicated section on the limitations of the study, could enhance the overall quality and impact of the research.

Some correction:

Simple summary

L11 – curves are scarce

L11 - lactation curves and investigates the influencing

Introduction

L35 – please add Najari et al. before the [7] in the beginning of the sentence

L36 – change aptation for adaptation

L37 – change capacicty for capacity

L38 – remove the extra space between selection and reduce

L39 – change whith for with

L39-41 – add a reference for such affirmation

L44 – delete the repeated word “the”

L44 – change comptabilised for compatibilized

L60 – change costsaving for cost saving

L65 – add a space after the citation.

L75 – change managment for management

L76 – change this for these

L78 – change essentaul tu for essential to

L79 - change this for these

Material and Methods

L92 – Add space after the dot

L133 – change capacitie for capacity

Table 1- please display the herds by ascendant order (1,2,3,4,5), also change herd 8 for 5

L142 – add a space between et al. and [29]

L148 – remove the space between production and ;

L149 - remove the space between production and ;

L154 – replace to resolve for to solve

L156 – remove the space between production and :

L157 – explain what is the +7 on the end of the sentence

L158 – remove the space between length and :

L162 – change non genetic for non-genetic

L169 – delete the space between follows and :

Equation 2: use a different letter to denote the year of kidding effect ( Y is already been used to represent the estimated curve parameter, this can cause some confounding).

Results:

L181- remove repeated word the

Table 2:  what is (g/j)?

L190 - remove space between SD and :

L192 – change ptoductive for productive

L193 – correct ”evolutedun der”

L194 – change adults for adult

L196 – Explain what is g/j

L196 – add the unit after 0.26

L197-199 – add the unite immediately after the numbers

L246 – change pregnants to pregnant

Table 4 – add legend for the names using in the columns

L255 – change there were for there was a

L274 – change ptoductive for productive

L275 – correct evolutedun

L284 – correct mentionned

L305 – delete the extra space

L300 – change goats population for goat population

L305 – correct nomadisme

L310 – correct longterm

L313 – change reflects for reflect

L312 – correct mentionned

L324 – Add a space before Similar

L339 – correct gots

L341 – add space before examined

L344 – remove repeated word the

L360 – remove extra space between Morever and Devendra

L364 – add space between et al and [18]

Conclusion

L367 – correct analyzind

L368 – correct paramerters

L377 – correct tha kids

L378 – correct abjective

Comments on the Quality of English Language

Conducting a thorough English review would enhance the manuscript's clarity and coherence

Author Response

(The authors gave the same response as above.)

Round 2

Reviewer 2 Report

Comments and Suggestions for Authors

The necessary correction were made. Thank your for the improvements.

Author Response

Thank you so much for you revision . 

Kind regards 

Reviewer 3 Report

Comments and Suggestions for Authors

L129 – add a “)” after the citation

L161 – delete extra numbers after the “.”

L165 – change “To fix” for “to fix”

Please standardize the terms, Table 2 you call the model as wood but after that you refer as gamma, maybe using the term wood gamma would be better for all manuscript, please modified in the entire text.

L277 – the term “goats pregnant twin pregnant goats” doesn’t read well, perhaps change for  “goats that had twins pregnancy”  

Author Response

Thank you sincerely for your positive feedback and constructive evaluation of our article. Below, you will find responses addressing these considerations and the corresponding modifications made in the revised manuscript. Thank you for your time and thoughtful review.
